# VISIONARY-R1: MITIGATING SHORTCUTS IN VISUAL REASONING WITH REINFORCEMENT LEARNING

## ABSTRACT

Learning general-purpose reasoning capabilities has long been a challenging problem in AI. Recent research in large language models (LLMs), such as DeepSeek-R1, has shown that reinforcement learning techniques like GRPO can enable pre-trained LLMs to develop reasoning capabilities using simple question-answer pairs. In this paper, we aim to train visual language models (VLMs) to perform reasoning on image data through reinforcement learning and visual question-answer pairs, without any explicit chain-of-thought (CoT) supervision. Our findings indicate that simply applying reinforcement learning to a VLM—by prompting the model to produce a reasoning chain before providing an answer—can lead the model to develop shortcuts from easy questions, thereby reducing its ability to generalize across unseen data distributions. We argue that the key to mitigating shortcut learning is to encourage the model to interpret images prior to reasoning. Therefore, we train the model to adhere to a caption-reason-answer output format: initially generating a detailed caption for an image, followed by constructing an extensive reasoning chain. When trained on 273K CoT-free visual question-answer pairs and using only reinforcement learning, our model, named Visionary-R1, outperforms strong multimodal models, such as GPT-4o, Claude3.5-Sonnet, and Gemini-1.5-Pro, on multiple visual reasoning benchmarks. Code and models will be publicly released.

## 1 INTRODUCTION

Reasoning is essential for enabling AI to tackle complex problems and make informed decisions in real-world applications. However, training AI models to reason is extremely challenging—primarily due to the lack of large-scale human-annotated reasoning data (Lightman et al., 2023; Christiano et al., 2017; Ouyang et al., 2022). Recent advances in large language models (LLMs), such as DeepSeek-R1 (Guo et al., 2025a), have demonstrated the potential to induce reasoning capabilities in LLMs via reinforcement learning and using only question-answer pairs, without explicit step-by-step supervision. Meanwhile, the computer vision community has begun exploring RL approaches for visual language models (VLMs), using methods like GRPO (Shao et al., 2024) to extend reasoning to multimodal settings (Meng et al., 2025; Feng et al., 2025; Liu et al., 2025; Shen et al., 2025). While these efforts are promising, existing visual reasoning models often rely on complex multi-stage training pipelines that are both computationally expensive and time-consuming. Moreover, these models heavily rely on labeled chain-of-thought reasoning data distilled from proprietary models like GPT-4o—limiting scalability and openness.

In this paper, we aim to lower the development cost of training VLMs for visual reasoning by using only reinforcement learning and paired visual question-answer data, *without relying on any chain-of-thought supervision*. Inspired by DeepSeek-R1, we adapt GRPO to training VLMs using only question-answer pairs. Specifically, given an image and a question, we prompt a VLM to generate a reasoning chain followed by an answer and optimize the model using a combination of an accuracy reward (that evaluates the answer correctness) and a format reward (that encourages the reason-answer output format). However, this seemingly straightforward setup leads to a critical failure mode: the model develops *shortcuts* by producing short, uninformative reasoning chains. These shortcuts often suffice to answer easy training questions correctly, but the model fails to generalize to harder questions that require genuine visual understanding. As illustrated in Fig. 1, the model trained with GRPO performs well on simple training examples by exploiting shortcuts (top), but at test time, it produces incoherent reasoning and incorrect answers on unseen examples (bottom).

**Easy Sample (Train)**

**Q: How many hospitals in Colorado were owned by the state or local government in 2019? A: 26**

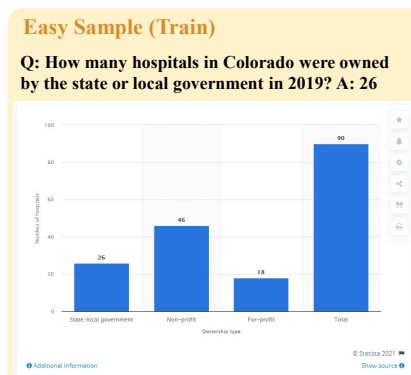

**GRPO**

\<think\> The number of hospitals in Colorado owned by the state or local government in 2019 is 26.\</think\> \<answer\> 26 \</answer\>

**Visionary-R1**

\<info\> The image is a bar graph that displays the number of hospitals in Colorado by ownership type. Here are the key points:
- **The total number of hospitals in Colorado is 90.**
- **26 hospitals are owned by the state/local government.**
- **46 hospitals are non-profit.**
- **18 hospitals are for-profit.**
\</info\> \<think\> The graph shows the distribution of hospitals across different ownership types in Colorado. **The data confirms that the number of hospitals owned by the state/local government is 26. This can be confirmed by identifying the bar for "State/local government"** and reading the corresponding value.
\</think\> \<answer\> 26 \</answer\>

---

**Hard Sample (Test)**  **Q:How many queries have a p-value lower than 0.50? A:4**

|        | shutdown | bullet | objective | superintelligent | rot23 | paperclips | paperclips_large | water |
|--------|----------|--------|-----------|------------------|-------|------------|------------------|-------|
| p-value | **0.92** | **0.69** | 0.01 | **0.10** | **0.58** | **0.43** | **0.60** | 0.01 |

Table 1: **Goodness-of-fit results for power law models.** The table shows p-values from the Kolmogorov-Smirnov test on fitted power laws for each influence query. Values above the 0.1 thresholds suggested by Clauset et al. (2009) indicate the power law is a plausible fit.

**GRPO**

\<think\> To answer the question, **we need to count the number of p-values that are lower than 0.50.** \</think\>
\<answer\> 3 \</answer\>

**Visionary-R1**

\<info\> The image contains a table titled "Table 1: Goodness-of-fit results for power law models." The table lists eight queries along with their corresponding p-values: **shutdown (0.92), bullet (0.69), objective (0.01), superintelligent (0.10), rot23 (0.58), paperclips (0.43), paperclips_large (0.60), and water (0.01).** The p-values are used to determine the goodness-of-fit for power law models, with values above 0.1 suggested by Clauset et al. (2009) indicating a plausible fit. \</info\> \<think\> To determine how many queries have a p-value lower than 0.50, **we need to count the number of p-values that are less than 0.50. From the table, the p-values are: 0.92, 0.69, 0.01, 0.10, 0.58, 0.43, 0.60, and 0.01. The p-values that are less than 0.50 are 0.01, 0.01, 0.1, and 0.43.** Therefore, there are four queries with p-values lower than 0.50. \</think\> \<answer\> 4 \</answer\>

Figure 1: Comparison between the GRPO model and Visionary-R1. Using the reason-answer output format, the GRPO model tends to generate shortcut responses for easy samples during training, which hinders the model from learning general-purpose reasoning capabilities and results in poor generalization performance. In contrast, with a more comprehensive understanding of the image context, i.e., using the caption-reason-answer output format, Visionary-R1 consistently generates long, meaningful reasoning chains for both easy and hard samples.

To address the shortcut issue, we propose **Visionary-R1**, a reinforcement learning framework that enforces visual understanding before reasoning. The key idea is to train the model in a structured caption–reason–answer format, where it must first generate a detailed caption of the image before reasoning and answering. The captioning step ensures that the model does not just rely on superficial cues or patterns but engages in a deeper analysis of the image context, regardless of whether the question is easy or hard—this forces the model to adopt a consistent problem-solving approach, thus mitigating potential shortcuts and consequently making the reasoning capabilities more generalizable across different data distributions. To ensure the caption is informative, we impose auxiliary supervision on the caption tokens by using reinforcement learning from

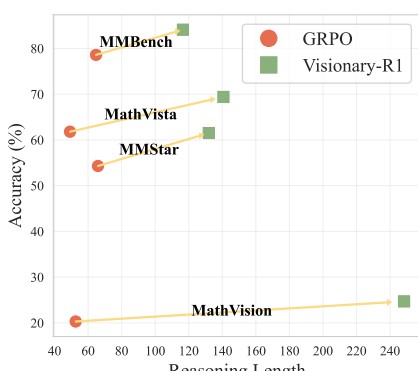

Figure 2: The longer the reasoning chain, the better the accuracy.

AI feedback (Bai et al., 2022). This caption reward is combined with standard accuracy and format rewards during policy optimization. The resulting model produces longer, more meaningful reasoning tokens than the model learned with GRPO alone (see Fig. 1), leading to better generalization performance on unseen data (see Fig. 2).

To evaluate our approach, we compile a comprehensive dataset that aggregates 11 popular question-answer datasets, covering areas such as scene understanding, chart analysis, mathematical problem-solving, and document processing. In total, the training data consists of 272.6K CoT-free question-answer pairs. After training, Visionary-R1 is evaluated on several challenging visual reasoning

benchmarks including MathVista (Lu et al., 2023), MathVision (Wang et al., 2024), MMBench (Liu et al., 2024), MMMUPro (Yue et al., 2024), MMStar (Chen et al., 2024a), and CV-Bench (Tong et al., 2024). The results show that Visionary-R1 outperforms strong proprietary models, such as GPT-4o, Claude3.5-Sonnet, and Gemini-1.5-Pro, as well as the latest competitors based on supervised pre-training and reinforcement fine-tuning.

In summary, we make the following contributions in this paper: 1) We share an important finding that GRPO does not work directly with VLMs due to shortcut learning; 2) We address the shortcut learning problem with Visionary-R1, a simple reinforcement learning-based model that interprets images before reasoning; 3) Through extensive experiments, we show that despite using only question-answer pairs, Visionary-R1 beats strong multimodal models, such as GPT-4o, Claude3.5-Sonnet, and Gemini-1.5-Pro, on challenging visual reasoning benchmarks. Code and models will be publicly released to facilitate future research.

## 2 RELATED WORK

**Supervised Learning for Visual Reasoning**    Learning LLMs/VLMs that can reason have gained increasing attention from both academia and industry due to their ability to generate human-like, step-by-step reasoning, which is advantageous for tackling complex problems and delivering more interpretable answers (Wei et al., 2022; Kojima et al., 2022). Supervised fine-tuning (SFT) is the most straightforward method to enhance a model's reasoning capabilities, which relies on labeled data containing thinking processes. Since collecting human annotations is costly, existing work often resorts to using a pre-trained model like OpenAI's GPT-4o to generate reasoning labels. For instance, LLaVA-CoT (Xu et al., 2024) utilizes GPT-4o to label 100K visual question-answer datasets with detailed chain-of-thought including summary, caption, and reasoning. However, the process of collecting CoT labels can be quite expensive, and the use of GPT-4o limits scalability while introducing a significant performance upper bound. Similarly, MMCR (Yan et al., 2025) also creates a 310k multi-turn reasoning dataset using GPT-4o. CoMCTS (Yao et al., 2024a) introduces the Mulberry-260k dataset, which is specifically crafted to train tree-structure reasoning models. Compared to these models, our Visionary-R1 only uses simple question-answer pairs for training *without any chain-of-thought supervision*, yet it achieves stronger reasoning performance.

**Reinforcement Learning for Visual Reasoning**    Compared to SFT, reinforcement learning (RL) has recently been proved more effective in developing general-purpose reasoning capabilities as this paradigm has the potential to enable the model to explore reasoning in a broader language space and develop its own thinking processes (Chu et al., 2025). Insight-V (Dong et al., 2024) presents a multi-agent system to select preference data from self-generated reasoning paths and optimizes the model based on a preference learning algorithm. R1-VL (Zhang et al., 2025) designs step-wise rewards to improve reasoning accuracy and validity but relies on labeled data for SFT. RL has also been applied in Vision-R1 (Huang et al., 2025) and R1-Onevision (Yang et al., 2025), but only 10K samples are used in these models for RL training while the main focus is on SFT (that uses more than 200K samples). Similarly, the Pixel Reasoner (Su et al., 2025) and VL-Rethinker (Wang et al., 2025) encourage deeper reasoning through images or explicit textual self-reflection, but their training pipelines still heavily rely on SFT with complex dataset selection and annotation processes. Our Visionary-R1 departs from the popular SFT-followed-by-RL pipeline and adopts a pure RL approach, eliminating reliance on large-scale annotated datasets required for SFT and enables more flexible, autonomous reasoning through RL-from-AI-feedback.

## 3 METHODOLOGY

We propose **Visionary-R1**, a reinforcement learning framework designed to improve the reasoning capabilities of VLMs, which can be trained using only visual question-answer pairs *without any explicit CoT supervision*. In what follows, we first highlight the shortcut issue that arises when applying RL to visual reasoning tasks (Section 3.1), then introduce our Visionary-R1 framework, which train the model to follow the caption-reason-answer output format, i.e., first generating an informative caption to understand the image context, followed by an extensive reasoning chain.

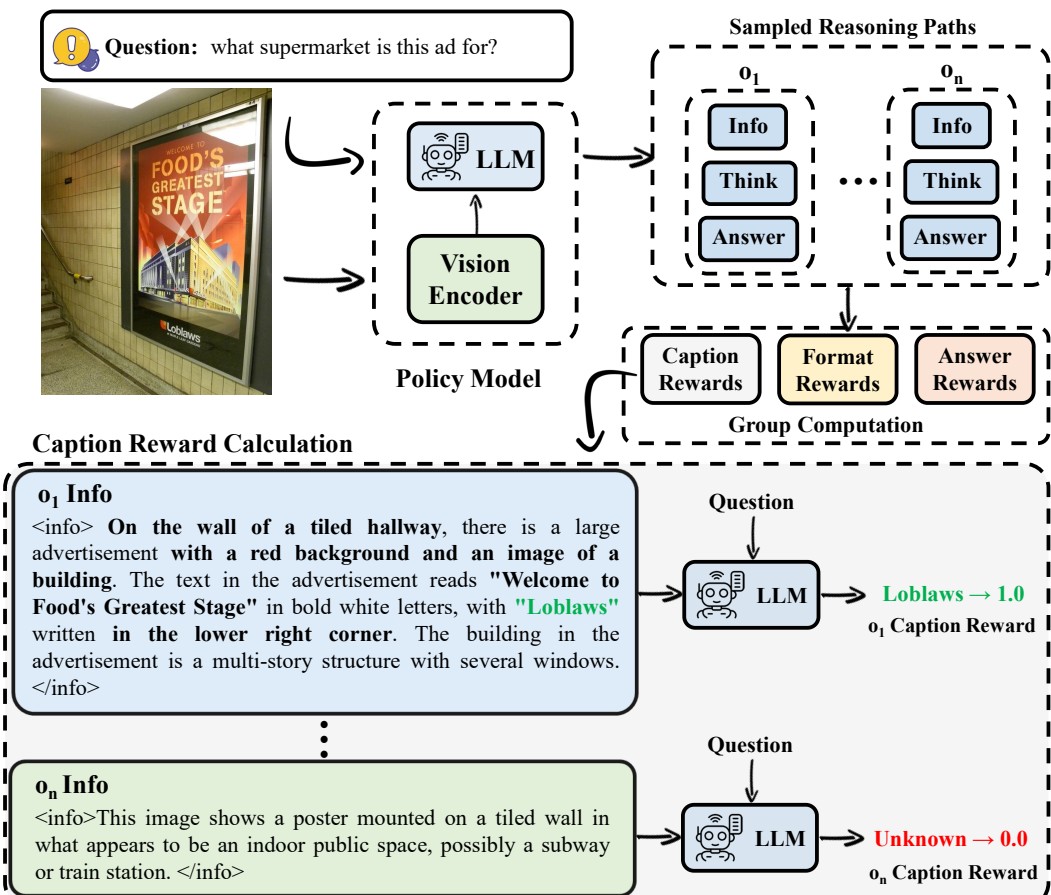

Figure 3: Overview of Visionary-R1. The primary training pipeline utilizes the GRPO method, which generates multiple reasoning paths for each question-answer pair. Additionally, an info tag is incorporated when calculating the format reward, and the policy model's LLM part is used to answer questions based on the description between the info tags, serving as the caption rewards. All rewards are then aggregated to determine the final advantage of each path.

## 3.1 MOTIVATION: THE SHORTCUT PHENOMENON IN VISUAL REASONING

While the GRPO (Shao et al., 2024) algorithm has been shown effective in improving the reasoning capabilities of language models, we observe a critical failure mode when transferring to visual reasoning tasks. This phenomenon manifests as a shortcut—GRPO often leads to degenerate behaviors where the model *ignores the visual input and relies primarily on textual patterns from the question to generate an answer*. As shown in Fig. 1, the model trained with GRPO can produce correct answers for simple questions during training—yet this is achieved without grounding in the image. This shortcut behavior can be particularly problematic in visual reasoning tasks, where the correct answer often depends on subtle image features such as embedded text, numerical values, object relationships, or chart patterns. Without forcing the model to attend to these visual signals, reinforcement learning alone encourages reward hacking: the model learns to exploit training distribution artifacts instead of learning general-purpose reasoning. To address this challenge, we propose a simple but effective modification: force the model to explicitly interpret the image before it begins reasoning. We operationalize this through a caption reward design (Section 3.2), which is then explicitly incorporated into the RL training objective (Section 3.3).

## 3.2 VISIONARY-R1: GROUNDING REASONING VIA CAPTIONING

**Caption-Reason-Answer Output Format**  We train the model to first generate captions before reasoning. This is operationalized via the caption-reason-answer output format:

1. **Caption**: generate a detailed description of the image, capturing objects, numbers, text, spatial relations, and other salient visual features;

2. **Reason**: construct a reasoning chain based on the captioned content;

3. **Answer**: provide the final answer to the question.

Specifically, we prompt the model to generate a detailed description, which is wrapped using a *<info></info>* tag. The final format we request the model to follow is therefore

*<info>...</info>*   *<think>...</think>*   *<answer>...</answer>*

The output is evaluated using a binary format reward $r_f \in \{0, 1\}$, which checks whether the generated response adheres to this format.

**Caption Reward**   While the format enforces structure, it does not guarantee that the caption is sufficiently detailed to support reasoning. To address this issue, we introduce a specialized caption reward $r_c \in \{0, 1\}$ based on reinforcement learning from AI feedback (Bai et al., 2022). Specifically, we feed the generated caption into an LLM, and ask it to answer the question based solely on the caption. In implementation, we use the LLM component of the policy model. If the answer is correct, the caption is deemed informative and rewarded; otherwise, it is penalized. This encourages the model to produce useful, visually grounded descriptions. The final reward for a sampled sequence $i$ is computed as:

$$R_i = r_a + r_f + \alpha r_c, \tag{1}$$

where $r_a$ is the accuracy reward and $\alpha$ is a balancing weight controlling the contribution of the caption reward.

### 3.3 TRAINING OBJECTIVE WITH CAPTION REWARD

Group Relative Policy Optimization, known as GRPO, was originally developed in DeepSeek-Math (Shao et al., 2024) for text-only reasoning tasks, and later adopted in DeepSeek-R1 (Guo et al., 2025a). GRPO simplifies the reinforcement learning paradigm by getting rid of the critic model. This is done by generating a group of responses for each sample and then computing the normalized reward within the group to determine an advantage value. To adapt it to visual reasoning, our method introduces two key differences. (1) First, as described in Section 3.2, we design a new reward structure by adding a caption reward that explicitly evaluates whether the model has interpreted the visual input, addressing the shortcut issue. (2) Second, we introduce a cosine-annealed KL penalty to stabilize training and encourage longer, more meaningful outputs—avoiding the limitations of a static KL coefficient in multimodal settings. We now detail our training objective and implementation.

**Policy Optimization**   For each training sample (i.e., a question-image pair), we sample $n$ response sequences $\{o_1, o_2, ..., o_n\}$ from an old policy model $\pi_{\theta_{\text{old}}}$. Each output is scored using the combined reward $R_i$ from Eq. 1. Then, an advantage value based on the $n$ rewards, $\mathbf{R} = \{R_1, R_2, ..., R_n\}$, is computed as

$$A_i = \frac{R_i - \text{mean}(\mathbf{R})}{\text{std}(\mathbf{R})}, \quad i = 1, \cdots, n. \tag{2}$$

The updated policy $\pi_\theta$ is trained using a clipped surrogate objective

$$\mathcal{J}(\theta) = \mathbb{E}[q \sim P(Q), \{o_i\}_{i=1}^n \sim \pi_{\theta_{\text{old}}}(O|q)]$$
$$\frac{1}{n} \sum_{i=1}^n \left( \min \left( \frac{\pi_\theta(o_i|q)}{\pi_{\theta_{\text{old}}}(o_i|q)} A_i, \text{clip} \left( \frac{\pi_\theta(o_i|q)}{\pi_{\theta_{\text{old}}}(o_i|q)}, 1 - \varepsilon, 1 + \varepsilon \right) A_i \right) - \beta D_{\text{KL}} \left( \pi_\theta \| \pi_{\text{ref}} \right) \right), \tag{3}$$

where both $\varepsilon$ and $\beta$ are hyper-parameters. $\varepsilon$ controls the clipping bound and limits the range of policy updates to avoid large changes that could destabilize training. $\beta$ is the KL penalty coefficient that regularizes deviation from a reference policy $\pi_{\text{ref}}$.

**Cosine Annealing KL Coefficient**    The KL penalty is formulated as

$$\mathbb{D}_{KL}\left[\pi_\theta||\pi_{ref}\right] = \frac{\pi_{ref}\left(o_i \mid q\right)}{\pi_\theta\left(o_i \mid q\right)} - \log\frac{\pi_{ref}\left(o_i \mid q\right)}{\pi_\theta\left(o_i \mid q\right)} - 1. \tag{4}$$

The KL divergence in Eq. 4 serves as a penalty term to prevent the model from straying too far from the baseline policy model, thereby stabilizing the training. It is non-trivial to determine the balancing weight for this term: using a large weight forces the model to stay within a close neighborhood of the baseline model and therefore impedes the model's ability to engage in more in-depth thinking and generating long, detailed reasoning; on the other hand, using a small weight can lead to unstable training and potentially result in reward hacking (Skalse et al., 2022). To overcome this challenge, we propose dynamically annealing the KL penalty coefficient over time using cosine annealing, which uses a large coefficient during the early, unstable training phase and gradually reduces the value to allow the model to produce longer outputs in later stages. Specifically, we replace $\beta$ in Eq. 3 with $\hat{\beta}$, which is calculated as

$$\hat{\beta} = \frac{\beta}{2} \times \left(1 + \cos\left(\pi \times \frac{T_{\text{cur}}}{T_{\text{max}}}\right)\right), \tag{5}$$

where $T_{\text{cur}}$ and $T_{\text{max}}$ represent the current and max training steps, respectively.

## 4 EXPERIMENTS

### 4.1 EXPERIMENTAL SETTINGS

**Training Data**    Unlike existing work that relies on curated data and reasoning labels, our approach allows the model to learn using CoT-free visual question-answer pairs. To ensure diversity, we aggregate 11 popular visual question-answer datasets by simply combining the training data without applying any preprocessing or filtering. The resulting training data consists of 272.6K visual question-answer pairs and covers a wide spectrum of visual formats, including general scenes, charts, tables, diagrams, math questions, documents, and 3D data. See Tab. 4 for details about the data composition.

**Benchmarks**    We evaluate our approach on several widely-used visual reasoning benchmarks that cover various visual formats and question types: MathVista (Testmini) (Lu et al., 2023), MathVision (Wang et al., 2024), and MMBench (en) (Liu et al., 2024). MathVista encompasses a variety of reasoning types, including logical, algebraic, and scientific reasoning questions. MathVision focuses on mathematical visual reasoning tasks. MMBench is a comprehensive evaluation suite concerned with visual and mathematical reasoning. Meanwhile, we also included the results of MMMUPro (Yue et al., 2024), MMStar (Chen et al., 2024a) and CV-Bench (Tong et al., 2024) in the A.4 to provide a more diverse and comprehensive evaluation.

**Baseline Methods**    To justify the effectiveness of our designs, we implement two baselines: 1) **SFT.** The model is directly trained with the original question-answer data. 2) **GRPO.** The model is trained with GRPO. These models are trained using the same backbone and training data as our approach. We also compare our approach with state-of-the-art methods reported in the literature, including both proprietary (e.g., GPT-4o, Claude3.5) and open-source models (e.g., InternVL2.5, LLaMA3.2).

**Implementation Details**    We adopt Qwen2.5-VL-3B (Bai et al., 2025) as the base model. This pre-trained model has strong visual understanding capabilities but has not undergone post-training for reasoning. For the group reward computation, we generate 8 output sequences (i.e., $n = 8$ in Eq. 3) and the sampling temperature is set to 0.9 following the common practice. All parameters are optimized with a learning rate of $5 \times 10^{-7}$. The caption reward's balancing weight $\alpha$ is set to 0.1. The KL coefficient $\beta$ is set to 0.04.

### 4.2 MAIN RESULTS

The results are shown in Tab. 1. Comparing SFT with the base model, we observe that the performance of SFT is worse on three out of four datasets, with the biggest performance decline reaching 12% on MathVision. These results suggest that the model learned with question-answer pairs overfits the

Table 1: Comparison with state-of-the-arts on three challenging visual reasoning benchmarks. SFT and RL mean supervised fine-tuning and reinforcement learning, respectively. CoT means chain-of-thought, which is either self-generated or distilled from third-party models like GPT-4o. QA means that the model is learned with question-answer pairs only. Despite having only 3B parameters and using only QA data for training, Visionary-R1 beats strong commercial models like GPT-4o and Claude3.5-Sonnet. Note that * indicates results borrowed from the Seed's report (Guo et al., 2025b).

|  | Size | Strategy | Data | MathVista | MathVision | MMBench |
|---|---|---|---|---|---|---|
| *Close-source models* | | | | | | |
| GPT-4o* (Hurst et al., 2024) | - | - | - | 63.8 | 31.2 | 84.3 |
| GPT-o1* (Jaech et al., 2024) | - | - | - | 71.8 | 63.2 | 83.8 |
| Claude3.5-Sonnet (Anthropic, 2024) | - | - | - | 67.7 | 37.9 | 82.6 |
| Claude3.7-Sonnet* (Anthropic, 2025) | - | - | - | 74.5 | 58.6 | 82.0 |
| Gemini-1.5-Pro (Team et al., 2024) | - | - | - | 63.9 | 19.2 | 73.9 |
| Gemini-2.5-Pro* (Google, 2025) | - | - | - | 82.7 | 73.3 | 90.1 |
| *Open-source models* | | | | | | |
| Qwen2.5-VL (Bai et al., 2025) | 3B | - | - | 62.3 | 21.2 | 79.1 |
| InternVL2.5 (Chen et al., 2024b) | 4B | - | - | 60.5 | 20.9 | 81.1 |
| MiniCPM-V2.6 (Yao et al., 2024b) | 8B | - | - | 60.6 | 17.5 | 81.5 |
| LLaMA3.2 (AI, 2024) | 11B | - | - | 51.5 | - | 65.8 |
| *Reasoning models* | | | | | | |
| Ovis (Yan et al., 2025) | 4B | SFT | CoT | 66.6 | - | 79.3 |
| Mulberry (Yao et al., 2024a) | 7B | SFT | CoT | 63.1 | - | - |
| R1-Onevision (Yang et al., 2025) | 7B | SFT+RL | CoT | 64.1 | 29.9 | - |
| Insight-V (Dong et al., 2024) | 7B | SFT+RL | CoT | 59.9 | - | 82.3 |
| R1-VL (Zhang et al., 2025) | 7B | SFT+RL | CoT | 63.5 | 24.7 | - |
| LLaVA-CoT (Xu et al., 2024) | 11B | SFT | CoT | 54.8 | - | 75 |
| *Our models* | | | | | | |
| Base Model | 3B | - | - | 61.5 | 19.1 | 82.1 |
| SFT | 3B | SFT | QA | 54.6 | 7.0 | 80.7 |
| GRPO | 3B | RL | QA | 61.8 | 20.3 | 78.6 |
| Visionary-R1 | 3B | RL | QA | 69.4 | 24.7 | 84.1 |

training data distribution. GRPO slightly outperforms the base model, achieving improvements of 0.3% on MathVista, 1.2% on MathVision. However, GRPO underperforms the base model by 1.5% on MMBench, which suggests that visual reasoning is difficult to learn from just question-answer pairs. By digging into the outputs, we observe that GRPO often leads to shortcuts in easy training samples while produces short, useless reasoning answers for unseen samples, as illustrated in Fig. 1.

Compared to SFT and GRPO, Visionary-R1 demonstrates huge potential in learning general-purpose reasoning capabilities, evidenced by the improvements of 7.9% on MathVista, 5.6% on MathVision, and 2% on MMBench, over the base model. Compared with reasoning models that rely on labeled reasoning data, Visionary-R1 still maintains clear advantages on most datasets, despite using only question-answer pairs. Notably, Visionary-R1 even surpasses strong commercial AI models, such as GPT-4o, Claude3.5-Sonnet, and Gemini-1.5-Pro, on MathVista, and MMBench. These results strongly justify the effectiveness of learning to caption before reasoning.

### 4.3 ABLATION STUDY AND ANALYSES

**Effectiveness of Captioning and Caption Reward** We conduct an ablation study to evaluate the effectiveness of each component in Visionary-R1. Specifically, we start from the GRPO model and incrementally add the caption output format and the caption reward $r_c$. Instead of using the compiled 272.6K training data, we use individual datasets to save computation. Specifically, we perform two sets of experiments on different types of datasets (to ensure diversity): 1) training on ChartQA and testing on MathVista and MathVision, and 2) training on A-OKVQA and testing on MMStar and

**Test Data**

**Boxes per shipment**

| Stem | Leaf |
|------|------|
| 1 | 9 |
| 2 | 6 |
| 3 | 4 9 |
| 4 | 1 9 |
| 5 | 0 3 9 |
| 6 | 4 8 |

Q:A shipping company keeps track of the number of boxes in each shipment they send out. How many shipments had exactly 56 boxes? A:0

**GRPO**

<think> To find how many shipments had exactly 56 boxes, we need to look for the number 56 in the plot. </think>
**<answer> 1 </answer>**

**GRPO+Caption**

<info> The image is a stem-and-leaf plot representing the number of boxes in each shipment. The stem represents the tens digit, and the leaf represents the ones digit. </info> <think> To find the number of shipments with exactly 56 boxes, **we need to look at the row where the stem is 5 and the leaf is 6.** So, there is no entry in the row where the stem is 5 and the leaf is 6, which means there is no shipment with exactly 56 boxes. </think> **<answer> 0 </answer>**

**GRPO+Caption+$r_c$**

<info> The image is a stem-and-leaf plot representing the number of boxes in each shipment. The stem represents the tens digit, and the leaf represents the ones digit. **The plot shows the following data: 19, 26, 34, 39, 41, 49, 50, 53, 59, 64, 68.** </info> <think> To find how many shipments had exactly 56 boxes, we need to look for the number 56 in the plot. **In the stem-and-leaf plot, the number 56 would be represented by the stem 5 and the leaf 6. We can see that 56 is not listed in the plot.** Therefore, there are no shipments with exactly 56 boxes. </think> **<answer> 0 </answer>**

Figure 4: Visualization of different model outputs. The caption output format enhances the reasoning while the caption reward further makes the reasoning more in-depth by improving the caption quality.

Table 2: Ablation study on different components in Visionary-R1.

| Method | Train: ChartQA | | Train: A-OKVQA | |
|--------|-----------|------------|--------|----------|
| | MathVista | MathVision | MMStar | MMBench |
| Zero-shot | 61.5 | 19.1 | 52.4 | 82.1 |
| GRPO | 59.0 | 18.2 | 54.2 | 82.6 |
| GRPO+Caption | 62.6 | 20.9 | 60.4 | 85.5 |
| GRPO+Caption+Length Reward | 62.0 | 20.3 | 59.6 | 85.2 |
| GRPO+Caption+Caption Reward | **64.6** | **22.7** | **62.9** | **87.6** |

MMBench. Tab. 2 shows the results, which clearly demonstrate the effectiveness of the caption output and the caption reward. Fig. 4 further illustrates the differences in the outputs of different models. We also experiment with a simple length reward to encourage more detailed captions, but find that this superficial approach merely increases redundancy and reduces model performance. This further highlights the effectiveness of our caption reward strategy.

**KL Coefficient** We experiment with different strategies for selecting the KL coefficient $\beta$. Specifically, we evaluate the following designs: 1) static values, 2) linear decay, and 3) cosine annealing (proposed in Eq. 5). For static values, we choose 0.04 and 0.008: the former is a common practice while the latter is a smaller value for testing the effect. The results are reported in Table 3. We find that using a static value leads to the worst results while linear decay achieves significant improvement—this highlights the importance of using a dynamic KL coefficient during training. Cosine annealing performs slightly better than linear decay. We also apply the cosine annealing strategy to GRPO but observe no performance gain, which suggests that this design mainly affects the captioning component in Visionary-R1.

To better understand why the KL coefficient makes such a huge impact, we dig into several key metrics logged during training, i.e., output length, the format reward, and the caption reward. The full training processes are shown in Fig. 5 (top). When setting the KL coefficient to 0.04, which has been widely adopted as a standard practice in the literature, the output length rapidly climbs up and reaches an unreasonably high value at around 700 steps, and then falls back to the normal level at 100 tokens; meanwhile, both the format reward and caption reward decline drastically as the output length shoots up to an abnormal value, meaning that the model has collapsed in the middle of training. The model collapse is more clear in Fig. 5 (bottom): the model generates long but completely meaningless reasoning tokens. When using a smaller value of 0.008, we encounter the reward hacking issue (Stiennon et al., 2020): the model mistakenly generates a short reasoning chain

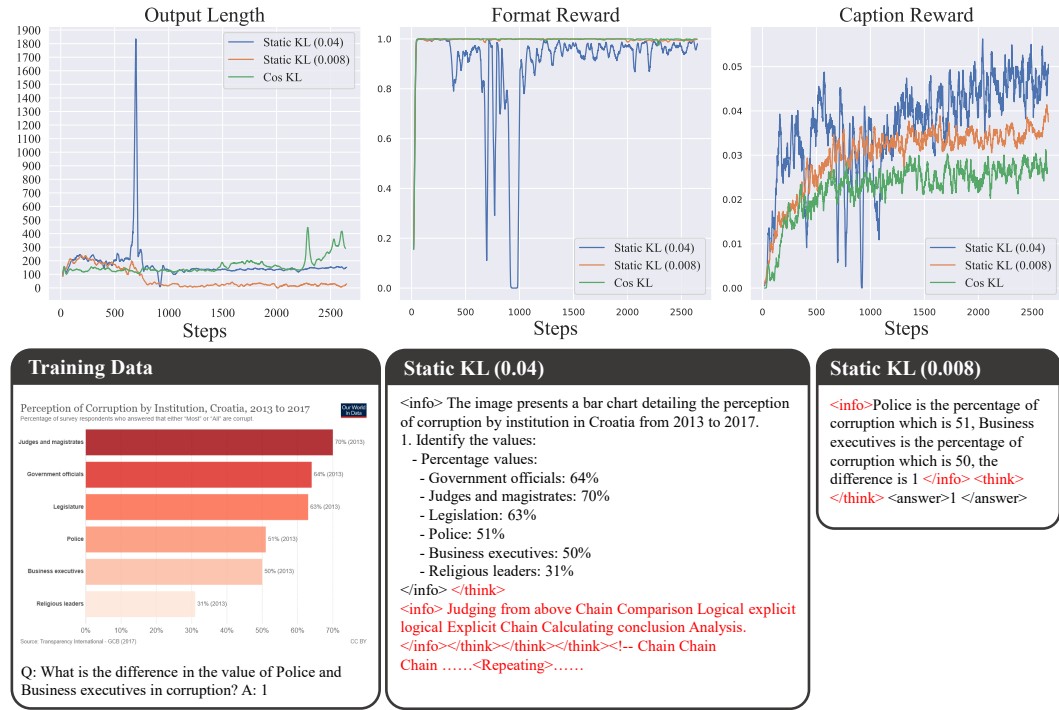

Figure 5: Visualization of curves for different KL coefficients (top) and output examples (bottom).

Table 3: Results of using different KL coefficients. Dynamic strategies (i.e., linear decay and cosine annealing) achieve significantly better results, with cosine annealing being the optimal choice.

| Method | Strategy | MathVista | MathVision | MMStar | MMBench |
|---|---|---|---|---|---|
| Visionary-R1 | Static (0.04) | 60.9 | 19.3 | 54.2 | 82.6 |
| | Static (0.008) | 60.7 | 18.7 | 56.0 | 82.7 |
| | Linear | 63.4 | 22.4 | 60.4 | 84.6 |
| | Cosine | 64.6 | 22.7 | 61.6 | 85.5 |
| GRPO | Static (0.04) | 59.0 | 18.2 | 48.1 | 80.4 |
| | Cosine | 59.6 | 18.4 | 46.6 | 80.9 |

at the caption place (which is supposed to contain a description about the image) while producing zero token in between *<think></think>*. This suggests that the model cheats in order to gain a higher accuracy reward and as a result the reasoning capabilities are not generalizable. The use of either linear decay or cosine annealing can effectively alleviate this issue.

## 5  CONCLUSION AND FUTURE WORK

This paper reveals the shortcut learning problem encountered when applying RL to VLMs. Unlike LLMs, VLMs are more difficult to train for reasoning without using annotated data. Visionary-R1, despite using CoT-free question-answer pairs, demonstrates strong performance on challenging visual reasoning benchmarks, surpassing strong commercial AI models that mostly likely benefit from larger-scale, higher-quality training data. The results indicate that understanding image context through captioning is essential for enhancing reasoning for VLMs. Moreover, the results also highlight the importance of the KL coefficient, which should be dynamically tuned to stabilize RL training. We believe the finding of the cosine annealing strategy could be applied more broadly to other RL applications. We believe that the effectiveness of RL training can be significantly amplified by using larger models. Investigation on larger-scale models is left as future work.

## ETHICS STATEMENT

We have carefully reviewed the ICLR Code of Ethics throughout this research. Our work does not involve human subjects, sensitive data, or practices that raise ethical concerns such as privacy, security, bias, or legal compliance issues. All the authors confirm that there are no conflicts of interest or other ethical issues related to this submission.

## REPRODUCIBILITY STATEMENT

We have made extensive efforts to ensure the reproducibility of our results. All details of our model architectures, training details, and hyper-parameters are described in Sec. 4.1 of the main paper, and all the training data and detailed prompts are completely listed in the Appendix. The code used for our experiments, along with the trained models, will be publicly released.

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

## A  TECHNICAL APPENDICES AND SUPPLEMENTARY MATERIAL

### A.1  COMPLETE LIST OF TRAINING DATA

Tab. 4 shows the complete training data, which aggregates 11 popular question-answer datasets and covers a wide range of visual formats and tasks, e.g., A-OKVQA (Schwenk et al., 2022) and TextVQA (Singh et al., 2019) for general scene understanding, ChartQA (Masry et al., 2022) and RoBUT SQA (Zhao et al., 2023) for chart understanding, GeoQA+ (Cao & Xiao, 2022) for mathematical problem-solving, and DocVQA (Mathew et al., 2021) for document processing.

Table 4: Composition of our training data.

| Dataset | Size | Answer Type | Visual Format |
|---------|------|-------------|---------------|
| A-OKVQA (Schwenk et al., 2022) | 17.1K | Multi-choice | General Scene |
| ChartQA (Masry et al., 2022) | 28.3K | Open-text+Num | Chart |
| AI2D (Kembhavi et al., 2016) | 15.5K | Multi-choice | Diagram |
| ScienceQA (Lu et al., 2022a) | 6.2K | Multi-choice | Scene + Chart |
| GeoQA+ (Cao & Xiao, 2022) | 12.1K | Multi-choice | Math |
| DocVQA (Mathew et al., 2021) | 39.5K | Open-text | Document |
| CLEVR-Math (Lindström & Abraham, 2022) | 32.6K | Num | 3D |
| Icon-QA (Lu et al., 2021) | 29.9K | Multi-choice | Diagram |
| TabMWP (Lu et al., 2022b) | 23.1K | Open-text+Num | Table |
| RoBUT SQA (Zhao et al., 2023) | 34.1K | Open-text+Num | Chart |
| TextVQA (Singh et al., 2019) | 34.6K | Multi-choice | General Scene |
| Total | 272.6K | | |

## A.2 POLICY MODEL PROMPT

To ensure the model interprets the image before engaging in the thought process, we include additional instructions in the system prompt to guide the policy model in generating the corresponding output. The complete model prompt can be seen from Fig. 6. Using this prompt, the model will insert the corresponding image description labeled as *<info>* before the thinking process, additional to the existing *<think>* and *<answer>*.

---

**Policy Model Prompt**

You are tasked with analyzing an image to generate an exhaustive and detailed description. Your goal is to extract and describe all possible information from the image, including but not limited to objects, numbers, text, and the relationships between these elements. The description should be as fine and detailed as possible, capturing every nuance. After generating the detailed description, you need to analyze it and provide step-by-step detailed reasoning for the given question based on the information. Finally, provide a single word or phrase answer to the question. The description, reasoning process and answer are enclosed within <info> </info>, <think> </think> and <answer> </answer> tags, respectively, i.e., <info> image description here </info> <think> reasoning process here </think> <answer> answer here </answer>.

---

Figure 6: System prompt given to the policy model.

## A.3 CAPTION REWARD PROMPT

Leveraging the language model within the policy model, we judge the level of detail by having the model answer questions based on the caption. A sufficiently detailed description of the image in the caption is essential for providing the necessary information to answer the questions accurately. With this approach, we prompt the language model to respond to questions based on the caption. To prevent reward hacking—where the model might include its thought process and answer in the information section—we incorporate an additional filtering command in the prompt to eliminate such interference. The complete caption reward prompt can be seen from Fig. 7.

---

**Caption Reward Prompt**

You are an analytical assistant designed to evaluate texts and answer questions based on strict criteria. Follow these steps:
Analyze the Text: Check if the provided text contains answers, solutions, explanations, problem-solving, or interpretations (e.g., reasoning steps, conclusions, causal statements like "because" or "therefore"). If any such elements exist, classify the text as non-descriptive.
Determine Response: If the text is purely descriptive (e.g., objectively describing images, diagrams, or scenes without explanations/answers), answer the user's question using only the description in a single word or phrase. If the text is non-descriptive, respond with "Hacking Sample".

---

Figure 7: System Prompt for the language model to answer the question based on the given caption.

## A.4 ADDITIONAL EXPERIMENTAL RESULTS

**Benchmark Evaluation**  Table 5 presents results on three challenging and diverse benchmarks: MMMUPro (Yue et al., 2024), MMStar (Chen et al., 2024a), and CV-Bench (Tong et al., 2024). Unlike

Table 5: Comparison results on the additional three challenging visual benchmarks. Visionary-R1 achieves stable improvements across all datasets.

| Methods | Size | MMMUPro | MMStar | CV-Bench-2D | CV-Bench-3D |
|---------|------|---------|--------|-------------|-------------|
| Base Model | 7B | 42.5 | 48.0 | 69.8 | 54.2 |
| R1-VL | 7B | 29.1(-13.4) | 60.0(+12.0) | 67.2(-2.6) | 65.9(11.7) |
| Base Model | 7B | 38.3 | 63.9 | 74.1 | 72.6 |
| R1-Onevision | 7B | 21.9(-16.4) | 59.1(-4.8) | 34.2(-39.9) | 20.1(-52.5) |
| Base Model | 3B | 31.6 | 52.4 | 72.6 | 71.1 |
| Visionary-R1 | 3B | 34.0(+2.4) | 61.5(+9.1) | 74.4(+1.8) | 74.0(+2.9) |

prior methods, which often display inconsistent performance and even significant regressions on certain datasets, our approach, Visionary-R1, achieves consistent improvements across all benchmarks. This stability indicates that the model's learned reasoning ability extends beyond dataset-specific adaptations, reflecting a more general and dependable form of multimodal reasoning.

**Model Scale Up Result** To evaluate the scalability of our approach in model scale, we conducted experiments using Qwen2.5-VL-7B as the base model and the A-OKVQA dataset (17.1K samples) for training. As shown in the Tab. 6, our method consistently outperforms the base model across all benchmarks for both the 3B and 7B model variants. These results provide strong evidence for the effectiveness and generalizability of our method.

Table 6: Experimental results of model scaling. The 17K data corresponds to training with the A-OKVQA dataset.

| Methods | Size | RL Data | MathVista | MathVision | MMStar | MMBench |
|---------|------|---------|-----------|------------|--------|---------|
| Base Model | 3B | - | 61.5 | 19.1 | 52.4 | 82.1 |
| Visionary-R1 | 3B | 17K | 62.5 | 20.5 | 62.9 | 87.6 |
| Base Model | 7B | - | 68.1 | 22.5 | 63.2 | 83.9 |
| Visionary-R1 | 7B | 17K | 70.2 | 24.4 | 66.7 | 89.5 |

**Hyper-parameter Analysis** For the GRPO method, we simply followed the original recipe by setting equal weights for the accuracy and format rewards. For the caption reward, we tried 0.1 and 0.5 as the balancing weight. The results are shown in the Tab. 7. The conclusion is that there is no significant difference between these two sets of results. To maintain training stability, we chose 0.1 and recommend that practitioners also use this setting.

Table 7: Results of hyper-parameter analysis for the reward weight $\alpha$.

| Weight $\alpha$ | MathVista | MathVision | MMStar | MMBench |
|-----------------|-----------|------------|--------|---------|
| 0.1 | 63.5 | 20.7 | 60.4 | 82.1 |
| 0.5 | 63.1 | 20.8 | 60.4 | 82.5 |

A.5 THE USE OF LARGE LANGUAGE MODELS (LLMS)

No LLMs were used in any part of the research ideation, writing, or editing processes for this paper. All content was independently completed by the authors.

## A.6 VISUALIZATION OF THE VISIONARY-R1 OUTPUT

To illustrate the validity of Visionary-R1 across various visual formats, we visualized the output for each format individually. The results are presented in Fig. 8-14.

**Document**

**Visionary-R1**

<info> The image shows a black and white document that appears to be a invitation or informational flyer. The details of the seminar are as follows:

- Title: Third Day Seminar
- Topic: Epidemiology & Prevention of Cardiovascular Diseases
- Organizer: Council on Epidemiology of the American Heart Association
- Collaborator: The National Heart, Lung & Blood Institute
- Date: July 30 - August 12, 1977
- Location: Snow Lake Lodge, Mount Snow, Vermont

The flyer has a minimalistic design with a logo featuring a heart and a torch at the top-left corner, which could be associated with the American Heart Association, given their logo often resembles a flame or torch within a heart shape. </info> <think> Given the information on the document, the key aspect to focus on is the location provided, which is explicitly mentioned as "Snow Lake Lodge, Mount Snow, Vermont." </think> <answer> Snow Lake Lodge, Mount Snow, Vermont </answer>

Q: Where the seminar is conducted?
A: Snow Lake Lodge, Mount Snow, Vermont

Figure 8: Visualization of Visionary-R1 Output in Document Format.

**General Scene**

**Visionary-R1**

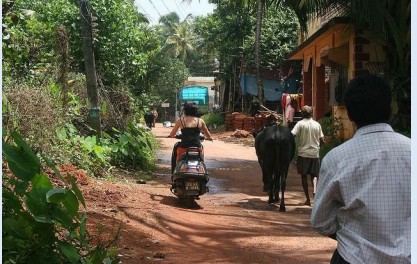

<info> In the image, there is a narrow dirt road in a rural setting, surrounded by lush greenery and tropical plants. A person is riding a motorcycle down the road, while another person is walking beside the motorcycle. There are also a cow is visible walking alongside the motorcycle. The street is appears to be in a village or countryside area. </info>
<think> The street in the image is dirt, indicating that it has not been paved. Paved streets are typically made from materials such as asphalt, concrete, or gravel, which are more resistant to weathering and wear. The lack of a paved surface suggests that this is a rural area where construction and maintenance of paved roads may not be as frequent or feasible. </think> <answer> no one </answer>

Q: Who paved this street?
A: no one

Figure 9: Visualization of Visionary-R1 Output in General Scene Format.

| Table | Visionary-R1 |
|---|---|
| tilapia — $4 per kilogram
catfish — $4 per kilogram
salmon — $2 per kilogram
halibut — $3 per kilogram
tuna — $4 per kilogram
trout — $3 per kilogram

Q: Manny wants to buy 3+9/10 kilograms of salmon. How much will he spend?
A: 7.8 | \<info> The image provides a list of fish prices per kilogram:
- Tilapia: $4 per kilogram
- Catfish: $4 per kilogram
- Salmon: $2 per kilogram
- Halibut: $3 per kilogram
- Tuna: $4 per kilogram
- Trout: $3 per kilogram
\</info>
\<think>
To calculate the total cost, we multiply the price per kilogram of salmon ($2) by the number of kilograms Manny wants to buy (3.9 kilograms).
The calculation is as follows:
Price per kilogram * Number of kilograms = Total cost
$2 * 3.9 = $7.80 \</think>
\<answer>
$7.80
\</answer> |

Figure 10: Visualization of Visionary-R1 Output in Table Format.

### 3D — Visionary-R1

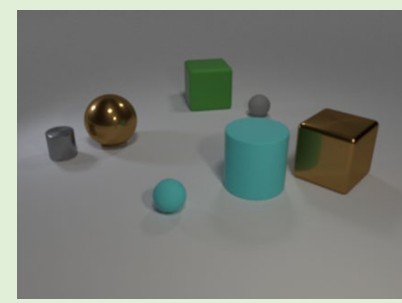

Q: Subtract all large green matte cubes. Subtract all big green blocks. How many objects are left?
A: 6

\<info> The image contains a variety of geometric shapes and objects. There is a large brown metallic sphere, a small gray metallic cylinder, a small gray matte sphere, a large green matte cube, a small cyan matte sphere, a large cyan matte cylinder, and a large brown metallic cube. \</info>
\<think> To determine how many objects are left after subtracting the large green matte cube and the big green block, we need to count the remaining objects. The large green matte cube is the same as the big green block, so we subtract one of each. This leaves us with the brown metallic sphere, the gray metallic cylinder, the small gray matte sphere, the cyan matte sphere, the cyan matte cylinder, and the brown metallic cube. Counting these, we find there are 6 objects left. \</think>
\<answer> 6 \</answer>

Figure 11: Visualization of Visionary-R1 Output in 3D Format.

### Chart — Visionary-R1

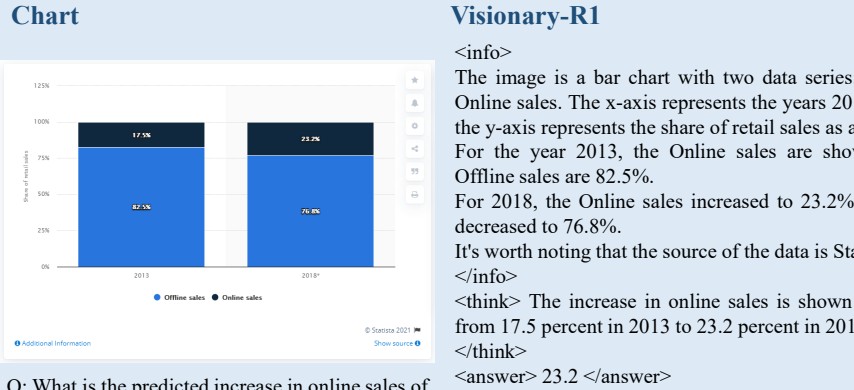

Q: What is the predicted increase in online sales of electricals and electronics by 2018?
A: 23.2

\<info>
The image is a bar chart with two data series: Offline sales and Online sales. The x-axis represents the years 2013 and 2018, while the y-axis represents the share of retail sales as a percentage.
For the year 2013, the Online sales are shown at 17.5%, and Offline sales are 82.5%.
For 2018, the Online sales increased to 23.2%, and Offline sales decreased to 76.8%.
It's worth noting that the source of the data is Statista 2021.
\</info>
\<think> The increase in online sales is shown in the percentage, from 17.5 percent in 2013 to 23.2 percent in 2018.
\</think>
\<answer> 23.2 \</answer>

Figure 12: Visualization of Visionary-R1 Output in Chart Format.

**Math**

**Visionary-R1**

Q: Place a ruler and a right-angled triangle ruler as shown in the figure (∠ACB is a right Angle). Given that ∠1=15°, what is the size of ∠2?
A: 45°

<info> A ruler and a right-angled triangle board. The right-angled triangle board has a 30° Angle. There are two marks on the ruler. One is in the middle and the other is in the lower left corner. The Angle indicated by the mark at the lower left corner is ∠2, which is calculated by the Angle between the right Angle of the right-angled triangle and the ruler. The Angle marked in the middle is ∠1. </info>
<think> According to the problem statement, one Angle of a right triangle is 30 degrees and the other Angle is 90 degrees. The problem states that ∠1=15 degrees. According to the formula for the sum of interior angles of a triangle, the other Angle (the Angle adjacent to the right triangle) is 60 degrees. Because the triangle set is a right triangle, the adjacent angles are 30 degrees. A right triangle has a right Angle of 90 degrees. So the Angle that ∠2 is adjacent to 90 degrees of a right triangle is 60 degrees minus 15 degrees, that is, 45 degrees. </think> <answer> 45° </answer>

Figure 13: Visualization of Visionary-R1 Output in Math Format. * *The original input and output were both in Chinese, and we have translated them directly without any modifications.*

**Diagram**

**Visionary-R1**

Q: Which is the top predator in the food web? Options: Birds, Crayfish, Fish, None of the above
A: Birds

<info> This image depicts a food web where organisms are connected by arrows indicating the direction of energy flow from one organism to another. The food web includes various organisms such as birds, fish, crayfish, worms, algae, and bacteria. The arrows show the feeding relationships among these organisms, with the arrows pointing from the prey to the predator. The food web forms a hierarchical structure where energy flows from the bottom to the top, starting from the bacteria and algae at the base and moving through various levels of consumers to the top predators. </info>
<think> To determine the top of the food chain, we need to identify the organisms that do not have any arrows pointing towards them. In this food web, birds are positioned at the top, with arrows pointing towards them from the organisms below. There are no arrows pointing towards birds from the bottom of the web. </think>
<answer> Birds </answer>

Figure 14: Visualization of Visionary-R1 Output in Diagram Format.

