# OpenReview forum: "Visionary-R1: Mitigating Shortcuts in Visual Reasoning with Reinforcement Learning"
_ICLR.cc/2026/Conference — ICLR 2026 Conference Withdrawn Submission_

### Official Review · Reviewer_F36n · 2025-10-19

**Soundness:** 3
**Presentation:** 3
**Contribution:** 2
**Rating:** 4
**Confidence:** 4

**Summary:**

This paper introduces Visionary-R1, a method to improve reasoning in multi-modal large language models. The authors show that traditional RL based approaches such as GRPO don't work well when applied to MLLMs because of shortcut reasoning, where the models apply short and un-informative reasoning traces to answer easy questions while training, leading to poor generalization and the models not reasoning well on OOD domains. To counter this, their approach makes the model follow a caption-reason-answer approach, where they incentivize the model to first produce a detailed caption of the image before producing the reasoning and answer sections. Their method also only utilizes end to end RL and does not use SFT unlike some of the existing work. On several multi-modal benchmarks, they show strong performance compared to baselines. They also show that adding a caption-reward and a cosine annealed KL divergance coefficient to their training recipe leads to improved performance.

**Strengths:**

1. The Visionary-R1 approach is simple, easy to scale and uses end to end reinforcement learning which makes it cheaper to train since it only uses QA pairs compared to some of the baselines which use SFT and hence need to obtain CoT traces.
2. The paper provides insightful ways to train VLMs and ablates the training recipe choices such as the caption reward and the annealed KL coefficient well, which can be useful for other practitioners training VLMs with RL.
3. Their 3B fine-tuned model beats several larger models on standard MLLM reasoning benchmarks by 4-5%

**Weaknesses:**

1. While the authors argue that training with GRPO directly leads to a shortcut reasoning issue, the paper lacks more detailed quantitative claims on this. It would be helpful if the authors can provide some metrics for example solution length over time when training with GRPO. Figure 5 does include some plots but since the KL coefficient is also being varied and it uses the caption reward, so it is hard to conclude. In addition to training metrics, it would be interesting to check this if the authors can provide the corresponding average solution length for their baselines (SFT, GRPO, Visionary-R1 ) for some the benchmarks they have presented in Table 1.
2. Adding on to point 1, the paper does not show how much the average solution length increases by when they use the caption + reason + answer paradigm. Since the captions can be quite long in certain cases, it is important to analyze the trade-off between larger solution lengths and accuracy, especially if some of the existing methods can achieve comparable performance with lesser number of tokens, it might question the need to perform the captioning step, especially for easier tasks.
3. The paper only covers performance on reasoning based benchmarks but does not talk about how the training generalizes to tasks which might not require reasoning. It is important to check if the end to end Visionary-R1 training paradigm leads to regressions tasks which do not have the <info><think><answer> format, like general multi-modal chat or text generation. One such benchmark which measures MLLMs holistically is MM-Vet (https://arxiv.org/abs/2308.02490). It might be useful if the authors can show results on this benchmark, especially the Language Generation subset. That should also help check if the model is able to generalize to different prompt formats not seen during training (like free flow chat questions)
4. The paper lacks further investigation on how performance scales with more data, and which of the 11 datasets used lead to the highest gains. Moreover, it is hard to compare some of the methods listed in the Table 1 without having the number of datapoints used for training of those baselines. It would be great if the authors can:
a. show data scaling curves (from ~25k - 272k) and how that leads to performance gains
b. a rough estimate of training data used for some of the reproducible open source baselines, which might allow for a fairer comparison between methods in Table 1
c. some analysis on which of the 11 training datasets leads to the largest gains

**Questions:**

Please refer to the weaknesses section. In addition:

1. Did you observe any reward hacking with the caption reward since you are using the same LLM for both assigning the caption reward and also as the policy model?
2. did you observe any hallucination issues with captioning which might lead to the model answering the questions incorrectly? Since the caption reward only takes into account whether the question can be answered with the caption, it does not explicitly prevent the model from hallucination in the captions.

---

### Official Review · Reviewer_j6io · 2025-10-29

**Soundness:** 2
**Presentation:** 3
**Contribution:** 1
**Rating:** 2
**Confidence:** 5

**Summary:**

Visionary-R1 is an RL framework based on GRPO that trains VLMs for visual reasoning using only question–answer pairs, without chain-of-thought labels. It enforces a caption-reason-answer structure and introduces a caption reward to ensure the visual description alone suffices for answering. Built on GRPO with a cosine-annealed KL penalty, the method improves performance of the base model on several benchmarks such as MathVista, MathVision, and MMBench.

**Strengths:**

1. Incorporating a caption-based reward into GRPO is an interesting extension of GRPO for multimodal reasoning.
2. The paper is clearly written and easy to follow.

**Weaknesses:**

[Major Weakness]
* The motivation for tackling the shortcut learning is not fully convincing. This issue might be mitigated by simply modifying the training data (e.g., filtering easy samples or injecting more complex QA pairs) without altering the training objective.
* The novelty is limited. Prior works such as [1] and [2] have already demonstrated that enforcing a model to caption before answering improves performance. As a result, it seems incremental because this work mainly enforces captioning before reasoning and answering.
* No discussion of inference efficiency is provided. Most of the gains reported over GRPO come from forcing the model to always produce captions and longer reasoning chains, even on trivial questions, which brings computational overhead.

[Minor Weakness]
* Enforcing caption generation uniformly might introduce unnecessary inference cost on easy questions that the VLM can already answer correctly. It might be more reasonable to conditionally trigger captioning only when the question is hard.

[1] Improving Visual Question Answering by Image Captioning (IEEE Access 2025)

[2] Enhancing Visual Question Answering through Question-Driven Image Captions as Prompts (CVPR 2024 Workshop on Prompting in Vision)

**Questions:**

1. The RL training seems to be applied without an SFT warm-up stage, unlike DeepSeek-R1. Please justify why pure RL is preferable or sufficient in this work.
2. In Table 1, Visionary-R1 does not outperform TBAC-VLR1. Any insights on the gap?

---

### Official Review · Reviewer_tjFT · 2025-10-30

**Soundness:** 2
**Presentation:** 2
**Contribution:** 3
**Rating:** 6
**Confidence:** 3

**Summary:**

This paper proposes the Visionary-R1 framework, which tackles the shortcut learning problem in vision-language models through reinforcement learning. The core innovation lies in introducing a “Describe–Reason–Answer” output format, forcing the model to first generate a detailed visual description before performing reasoning. Trained on only 273K visual question-answer pairs without reasoning-chain annotations, the model outperforms large commercial systems such as GPT-4o and Claude 3.5 on benchmarks like MathVista.

**Strengths:**

1. By enforcing the “Describe–Reason–Answer” output format, the model must first produce a detailed image description before reasoning. This clever design ensures the model deeply understands image content rather than relying on superficial patterns.

2. The dataset is broad, the evaluation benchmarks are comprehensive, and the paper provides extensive ablation studies and hyperparameter analyses, which strengthen the credibility of the conclusions.

**Weaknesses:**

1. The proposed approach still relies on a VLM to generate reasoning chains during training. How is this fundamentally different from methods that extract reasoning chains from proprietary models such as GPT-4o?

2. The discussion of related work is incomplete — the paper does not adequately cover other recent approaches to avoiding shortcut learning (e.g., causal intervention or disentangled representation learning) and lacks a thorough comparison with similar “intermediate supervision” methods (e.g., generating visual descriptions before reasoning).

3. Demonstrating the shortcut problem in GRPO only through the qualitative analysis in Figure 1 is insufficient. The authors should provide quantitative evidence, such as blind experiments, to confirm the existence of the issue.

**Questions:**

see the weaknesses

---

### Official Review · Reviewer_i5cF · 2025-11-01

**Soundness:** 2
**Presentation:** 3
**Contribution:** 2
**Rating:** 2
**Confidence:** 4

**Summary:**

The paper introduces Visionary-R1, a reinforcement learning (RL) framework for training visual-language models (VLMs) on visual reasoning tasks without any chain-of-thought (CoT) supervision. Inspired by DeepSeek-R1 and GRPO, the authors identify that direct RL fine-tuning on question–answer pairs causes shortcut learning, where the model overfits to easy samples by ignoring visual grounding. To mitigate this, Visionary-R1 enforces a caption–reason–answer output format, requiring the model to first describe the image (caption), then reason, then answer. The method introduces a caption reward (from AI feedback) to ensure informative visual grounding and applies cosine-annealed KL regularization to stabilize training.

**Strengths:**

1.The paper clearly identifies a practical failure mode — shortcut learning in visual RL — and provides strong empirical evidence of this phenomenon.
2. The caption–reason–answer structure and caption reward are conceptually simple but yield measurable generalization improvements.
3. The methodology is well-detailed, including architecture, rewards, and prompt templates. The authors commit to releasing code and models.

**Weaknesses:**

1. The caption reward relies on the model’s own LLM component to verify if the caption enables correct answering — this could lead to reward leakage or self-confirmation bias. There’s no analysis of how often the reward misfires.
2. While outperforming on MathVista/MMBench, the improvement on some datasets (e.g., MathVision, MMStar) is modest, suggesting the gains may stem from formatting or stylistic changes rather than deeper reasoning.
3. Longer outputs correlate with accuracy, but this metric may simply reflect verbosity rather than actual interpretive reasoning. There’s no human evaluation or visual attention analysis to confirm genuine grounding.
4. The paper claims to “mitigate shortcut learning,” but does not quantify shortcut severity reduction.
5. While the paper includes a few ablations (e.g., adding caption and caption reward), it does not disentangle all design contributions. Important components like the cosine-annealed KL penalty, format reward, or individual reward weights are not independently analyzed. Moreover, the ablations are limited to small subsets, making the conclusions less generalizable. The study also lacks statistical variance or multiple runs, so the reported gains may not be robust.
6. Missing zero-shot caption–reason baseline: The paper claims that enforcing a caption–reason–answer structure via RL mitigates shortcut learning, but it does not include a simple zero-shot or instruction-tuned baseline where the base model is merely prompted to follow the same format without RL fine-tuning. Such a baseline would clarify whether the improvement actually comes from the reinforcement learning signal or simply from the structured prompting itself. Without this comparison, the central claim—that Visionary-R1 develops reasoning rather than format bias—remains unproven.
7. Lack of quantitative evidence for grounding: The paper qualitatively argues that Visionary-R1 promotes visual grounding, but there is no metric measuring this (e.g., visual attention analysis, faithfulness, or caption relevance). The evaluation is purely based on accuracy, leaving it unclear whether the gains reflect genuine reasoning or just longer, well-structured responses.

**Questions:**

See Weaknesses

---

### Official Review · Reviewer_bsz4 · 2025-11-02

**Soundness:** 3
**Presentation:** 3
**Contribution:** 2
**Rating:** 4
**Confidence:** 5

**Summary:**

This paper aims to address the challenge of reasoning shortcuts in VLMs, where models often rely on textual priors instead of visual evidence. To overcome this, the authors propose Visionary R1, a visual reasoning–aware training framework that integrates GRPO with visual chain-of-thought supervision. The method encourages models to reason explicitly about visual cues and penalizes text-only heuristics, producing a more grounded reasoning process. Experiments across benchmarks demonstrate that Visionary R1 substantially reduces shortcut reliance and improves multimodal reasoning accuracy, outperforming both supervised fine-tuning and conventional RLHF approaches. The results highlight the effectiveness of R1-style visual reasoning training for mitigating spurious shortcut behavior in VLMs.

**Strengths:**

1)	Visionary-R1 introduces a conceptually simple but powerful “caption-before-reason” strategy that forces the model to understand the image context before reasoning.
2)	Unlike prior VLMs requiring large-scale GPT-4-generated CoT supervision, Visionary-R1 is trained purely from question–answer pairs, significantly improving scalability and autonomy.
3)	The inclusion of an auxiliary caption reward explicitly reduces the tendency to rely on superficial visual cues, encouraging deeper, generalizable reasoning.

**Weaknesses:**

1)	Because the R1 training process lacks explicit CoT supervision, it is uncertain whether the observed problems stem solely from shortcut learning. Other possible causes, such as unstable reward optimization or limited exploration, may also contribute. The paper should analyze these alternatives more carefully or justify why shortcut bias is the only plausible explanation.
2)	In Figure 1, the paper mentions “shortcut” but does not clearly define what it refers to. In this paper, it seems related to models exploiting textual cues instead of visual reasoning, yet the description lacks formal or measurable criteria. The authors should explicitly clarify what constitutes a shortcut and how it is identified or quantified.
3)	In line 92, the paper claims that the captioning step enables deeper image analysis rather than reliance on superficial cues. However, captioning typically focuses on describing visible, surface-level content rather than abstract reasoning. The paper does not clearly explain why or how generating captions contributes to improved reasoning ability. A stronger empirical or theoretical justification is needed to show that captioning truly enhances reasoning depth instead of merely restating image descriptions.
4)	Figure 2 shows that longer reasoning traces may improve performance, but some prior studies suggest that overthinking can harm efficiency and even accuracy [1]. The paper should clarify why increased reasoning length is considered beneficial here and whether there is evidence that such extended reasoning reflects genuine improvement rather than redundancy or noise. A discussion comparing “productive reasoning” versus “overthinking” would make the interpretation more balanced and convincing.
[1] More Thinking, Less Seeing? Assessing Amplified Hallucination in Multimodal Reasoning Models.
5)	The paper claims that Visionary-R1 mitigates shortcut reasoning and reduces hallucination, yet it is not evaluated on dedicated hallucination detection or grounding benchmarks. Testing on such datasets would provide stronger evidence that the method truly decreases visual hallucinations rather than merely improving task accuracy.
6.    Since the proposed Visionary-R1 framework introduces a new reinforcement training paradigm, open-sourcing the code is essential for reproducibility, fair comparison, and future research extensions.

**Questions:**

Please refer to the weaknesses.

---

### Note · Authors · 2025-11-13

I have read and agree with the venue's withdrawal policy on behalf of myself and my co-authors.